# Income Inequality and Obesity among US Adults 1999–2016: Does Sex Matter?

**DOI:** 10.3390/ijerph18137079

**Published:** 2021-07-02

**Authors:** Hossein Zare, Danielle D. Gaskin, Roland J. Thorpe

**Affiliations:** 1Department of Health Policy and Management, Johns Hopkins Bloomberg School of Public Health, Baltimore, MD 21205, USA; 2Global Health Services and Administration, University of Maryland Global Campus (UMGC), 624 N. Broadway, Hampton House 337, Baltimore, MD 21205, USA; 3Department of Population, Family and Reproductive Health, Johns Hopkins Bloomberg School of Public Health, Baltimore, MD 21205, USA; dgaskin5@jhu.edu; 4Program for Research on Men’s Health, Hopkins Center for Health Disparities Solutions, Johns Hopkins Bloomberg School of Public Health, Baltimore, MD 21205, USA; rthorpe@jhu.edu

**Keywords:** income inequality, obesity, Gini coefficient

## Abstract

Obesity is a major public health problem that varies by income and sex, yet there is little evidence to determine the association between income inequality and obesity. We examined the association between income and obesity in adults ages 20 years and older and tested whether this relationship differs by sex in the United States. We used the 1999–2016 National Health and Nutrition Examination Survey (NHANES). We defined obesity if the body mass index was ≥30 kg/m^2^, and calculated the Gini coefficient (GC) to measure income inequality by using the Poverty Income Ratio. We examined the association between income and obesity using a Modified Poisson regression in a sample of 36,665 adults. We adjusted the models according to age, racial/ethnic groups, marital status, education, health behaviors, health insurance, self-reported health, and household structure. The association between income and obesity was consistently more significant among middle-income quintile and higher-income quintile men than among lower-income quintile men. The same association was not found for women; women in the highest income quintile were less likely to suffer from obesity than among lower-income quintile women. Our results suggest that policymakers should consider strategies to reduce structural inequality and encourage access to healthy foods and community-supported agricultural programs as nutritional interventions in low-income population settings.

## 1. Introduction

America has experienced sharp increases in income inequality in recent decades. The Gini coefficient (GC)—a well-known index to measure income inequality—has increased almost every year, from 0.394 in 1974 to 0.462 in 2000, and 0.489 in 2017 [1]. In the last four decades, the real annual earnings for the top 1% and bottom 90% increased 158% and 24%, respectively. Despite a little progress, the income gap between men and women has also been demonstrated, where men earn higher wages than women; for example, in 2012 median earnings for full-time men ages 15 and older were $49,398 but $37,791 for women [2,3]. Unequal distribution of income may add an additional hazard to the health of people living in communities with unequally distributed income. This context increases the importance of understanding the relationship between income distribution and specific health conditions [4]. Studies have focused on an ecological association of income distribution and mortality as a main health outcome [5,6]. A negative association between income inequality and poor health outcomes has been documented in several studies [7,8].

The spread of income—income inequality—opens a new argument that the distribution of income may not have an equal impact on rich and poor communities because of the concavity effect of income inequality and health outcomes. In this approach, transferring an additional dollar from rich to poor improves the health status of poor communities more than rich communities [8]. With higher income inequality in society, we expect to have a higher proportion of people in poverty [9]. Income influences health outcomes by shaping health behaviors to reduce behavioral risk factors, decreasing the barriers to accessing care, moderating environmental factors, or reducing inequalities [10]. People with higher incomes tend to live in healthier neighborhoods and have higher educational attainment and more social capital [11]. Income has direct and indirect effects on the material conditions necessary for biological survival, social participation, and opportunity to control life circumstances [7]. Lower-income populations may have higher rates of behavioral risk factors including smoking, drinking, obesity, and lower levels of physical activity [2]. Additionally, sex plays a role in obesity; a few international studies showed that lower socioeconomic status increased the risk of becoming obese in women compared to that in men. [12,13] Despite a large body of knowledge regarding obesity and socioeconomic status, relatively little is known about income, income inequality, and its impact on obesity and its association with sex.

In 2017–2018, the prevalence of obesity—one of the well-known health outcome measures—was 42.4% in US adults; between 1999 and 2018, the age-adjusted prevalence of obesity increased from 30.5% to 42.4% [14] and the medical cost of adult obesity in the US was estimated from $147 billion to nearly $210 billion per year [15,16]. The studies showed that the prevalence of obesity fluctuated by income and socioeconomic status [17]. Studies have investigated the direct relationships between income and obesity [18,19,20]. For example, Subramanian et al. collected a large body of studies on the association between income inequality and health outcomes in the US and at international levels [20].

This association may operate differently by men and women. For instance, income inequality is negatively associated with weight status in men who are highly socially integrated, but is positively associated with weight status among men who have low social integration [21]. In spite of a negative correlation between income inequalities and health outcomes, evidence shows that Americans underestimate the true level of income inequality or may not be completely aware of its impact on health [22]; therefore, it is important to bring it to the attention of policymakers. The results of this study will inform policymakers about the need to address income differences and their impact on obesity. We investigated the relationship between obesity and income inequality and how this association changed between men and women between 1999 and 2016.

## 2. Materials and Methods

Data for this study came from the 1999–2016 National Health and Nutrition Examination Survey (NHANES) [23]. NHANES is a cross-sectional survey that provides nationally representative estimates of health and nutritional status for the US population, with a response rate of 73.2% between 1999 and 2016 [24,25], and a multistage probability sampling design that makes the sample representative of each of the four regions of the US [24]. The original sample between 1999 and 2016 was 42,584 individuals and 50.6% of the study population were female. For this study, we included participants who were 20 years old and older. We excluded pregnant women (1667) or missing observations for Poverty Income Ratio (PIR) (1823 men and 2429 women), which yielded an analytic sample of 18,518, men and 18,147 women.

### 2.1. Outcome Variable

Using body mass index (BMI)—derived by dividing weight in kilograms by height in meters squared (kg/m^2^)—we created a binary variable to identify participants who were obese (if BMI ≥ 30) as the outcome variable [26]. As stated by NHANES, “all body measures were obtained by trained health technicians in the body measures room for each of the Mobile Examination Centers” [27]. We need to note that both the high and low ends of the BMI spectrum are undesirable since high BMI is a risk factor for many chronic diseases and associated adverse outcomes. Low BMI can be an indicator for cancer or other severe poor health states that cause weight loss.

### 2.2. Main Independent Variable

The main independent variable of interest was income measured as the Poverty Income Ratio, the ratio of family income to poverty threshold. Using the income quintile approach, we defined a categorical variable with five quintiles from low to high.

Additionally, we calculated the Gini coefficient (GC) as a measure of income inequality to plot income inequality between obese and non-obese populations. The GC is a well-known single measure of inequality; the GC is based on the Lorenz curve. The Lorenz curve represents the actual distribution of income in a given society. The GC is defined as A/(A + B): A is the area between the line of perfect equality (45-degree line) and the Lorenz Curve; B is the area between Lorenz Curve x- and y-axis, if ‘A’ equal zero, then GC will be zero, which means perfect equality and if ‘B’ was zero then the GC will be one, which means complete inequality [28].

### 2.3. Covariate

For the demographic variables, we included age (years), racial/ethnic groups (White NH, Black NH, Mexican American, and other race) and marital status (1 = married, 0 = otherwise). For socioeconomic status (SES) variables, we included educational attainment (less than high school graduate, high school graduate or general equivalency diploma, more than high school education or some college and above) and having health insurance (1 = yes; 0 = no). To control for health status, we used self-reported general health status (excellent–very good, good–fair, and poor). Health behavior was measured by three variables: smoking (never smoked, former smoker and current smoker), drinking (never drink, former drinker and current drinker), and physical activity—a binary variable showed that an individual had not participated in vigorous activities (1 = yes; 0 = no) during a typical week. We also used a binary variable to present living alone (1 = yes; 0 = no) and female-headed household (1 = yes; 0 = no).

### 2.4. Analytic Strategy

For the first set of analysis, the mean and proportional differences between men and women for obesity, demographics, SES, health-related characteristics, and health behaviors were evaluated using unequal variances t-tests. In our sample, the prevalence of obesity was greater than 10%; therefore, we used a weighted modified Poisson regression analysis [29,30,31] that produced prevalence ratios (PR) and corresponding 95% confidence intervals (CI) [29,30]. We ran sets of adjusted and non-adjusted models (Models 1–6). Model 1 was an unadjusted model examining the relationship between poverty income ratio quintile and obesity for men. Model 2 was an adjusted Model 1 including sex (=1 if female) as a covariate. We also ran Model 3 with an interaction term of PIR quintile and sex. As the interaction between PIR quintile and sex was significant (*p* < 0.001), we stratified the analyses by sex. These resulted in Model 3 and Model 4 as unadjusted and adjusted models for men, and Model 5 and Model 6 for women, respectively.

To make our estimates representative of the national US civilian population, all analyses were weighted using the NHANES individual-level sampling weights for 1999–2016 (8 waves of data) [32]. We considered *p*-values <0.05 as statistically significant and all tests were two-sided. We used STATA statistical software version 15 to perform all analyses.

## 3. Results

### 3.1. Descriptive Analysis Results

Table 1 compares the distribution of the sample’s characteristics. On average, one-third of the sample was obese, with a significant higher rate of obesity for women (*p* < 0.001). Women’s obesity rates were lower than men’s only in the 5th PIR quintile (29.9% women vs. 32.8% men); otherwise, women experienced more obesity than men in the first, second, and third PIR quintiles. In Table A1, we compare the prevalence of obesity across all study characteristics (See Table A1).

Overall, the sample age was 47.0 ± 18.0 years with a slightly older population of women (47.9 [17.9] vs. 46.0 [18.0]). The women’s sample had a higher percentage of Black Non-Hispanics (NH) and the men’s sample had a higher percentage of Mexican Americans, with non-significant differences between the percentage of White NH and other racial/ethnic groups in both samples. The majority of women had more than a high school education (61.1%) and were covered by any kind of health insurance. Two-thirds of the women had never smoked (59.0%), 67.0% currently drank, 57.4% were physically active, and 82.5% were healthy. In comparison, 46.5% of men had never smoked, 85.2% currently drank, 61.5% were physically active, and 84.2% were healthy. More women lived alone than men (14.9% vs. 11.8%) and only 43.7% of the women were heads of households, compared to 72.3% of men. Table 1 details the information on men and women.

### 3.2. Association between Poverty Income Ratio Level and Obesity in Men and Women

The association between PIR levels and obesity in women and men is displayed in Table 2. The results of the unadjusted model show that people on the top quintile (5th quintile) were less obese (PR: 0.85; CI: 0.80–0.91) than people on the first quintile; however, the association disappeared in the adjusted model (PR: 0.94; CI: 0.87–1.01). The results of the adjusted model (Model 2) indicate that the obese population comprised more women, Black NH, or Mexican Americans. They had a higher probability of being married, having higher education, being former smokers and drinkers, not being physically active, and having poor health conditions.

### 3.3. Association between Poverty Income Ratio Levels and Obesity in Men

In previous sections, we reported the results for men and women, adjusted by sex, while considering the significant interaction between the PIR quintiles and sex, and we stratified the analyses for men and women. The association of PIR levels and obesity in the sample of men is displayed in Table 3. Models 3 and 4 present the association between PIR and obesity in men. There is a positive association between PIR and obesity in men in the unadjusted and adjusted models; for example, based on the unadjusted model, men in the 4th quintile (PR: 1.26; CI: 1.14–1.39) and 5th quintile (PR: 1.21; CI: 1.09–1.37) were obese. However, in the adjusted model, men in the middle (PR: 1.14; CI: 1.03–1.26) and in high PIR quintiles (PR: 1.20; CI: 1.09–1.34) were more likely than men in the lowest-income quintile to be obese. 

In Table 3, Models 5 and 6 present the association between the PIR and obesity in women. Based on the unadjusted model, women in the 3rd, 4th, and 5th PIR quintile group were less likely to be obese (PR: 0.92; CI: 0.85–0.99), (PR: 0.84; CI: 0.78–0.91) and (PR: 0.68; CI: 0.63–0.74), respectively. After adjusting the models, the association between PIR levels and obesity in the 4th quintile disappeared but remained significant in the very rich population (PR: 0.84; CI: 0.77–0.93). There is a negative association between the income inequality levels and obesity in women. Our adjusted models for men and women showed that individuals with obesity were Black NH or Mexican American, high school graduates, former smokers or drinkers, undertook no vigorous activities, and had fair–poor health conditions. In Figure 1, we compare the prevalence ratios between men and women in the final model (See Figure 1).

### 3.4. Obesity and Income Inequality in Men and Women by Estimating Gini Coefficients

As presented in the regression results, there is a different association between obesity and income inequality in men and women. To understand more about these differences in men and women, we used Lorenz curves and Gini coefficients. The Lorenz curves (Figure 2) show the Gini coefficient (GC) for the ‘income to poverty ratio’ in men’s and women’s populations in the US between 1999 and 2016. To plot these curves, we used the average GC with jackknife standard errors. Figure 2 compares the GC between men and women and between obese and non-obese populations by sex. In panel A, the blue solid line plots the distribution of the PIR in non-obese women and the dashed-red line plots the distribution in non-obese men. With GC 0.351 (SE: 0.002) and GC 0.348 (SE: 0.002) for women, we did not find any significant difference between non-obese men and women (*p* = 0.191), but there was a different pattern in the obese populations. For all quintiles, the red-dashed line is above the blue line, which means that obese women experienced higher income inequality than obese men (See panel B). For example, among obese men, lower than 25% of the population acquired only 7.1% of PIR and 53% of PIR was acquired by 75% of the population. The rest of the PIR was acquired by the top 25% of the population. Among obese women, these distributions changed to 6% and 48% of the first 25% and 75% of the population and 52% of PIR acquired by the top 25% of the population. The GC rose from 0.333 (SE: 0.002) in men to 0.377 (SE: 0.002) in women. The standard errors for these estimators were very small and there was a significant difference between the GC in men and in women (*p* < 0.001).

In panels C and D, we compare the GC between the populations of obese and non-obese men and women. As presented, obese women between the 5th and 10th percentile of the population suffered more from income inequality (panel C); the GC in obese women moved between 0.337 (SE: 0.003) to 0.351 (0.002) in non-obese women. For men, there is higher income inequality in the non-obese population (panel D). The red-dashed line that represents obese-men stays above the solid blue line (non-obese men), meaning lower income inequities for all income groups. The GC moved from 0.333 (SE: 0.003) in obese men to 0.348 (SE: 0.002) in non-obese men.

## 4. Discussion

In this study, we investigated the relationship between obesity and income measured by the PIR and how this association changes between men and women. For the first set of analysis and by calculating GC and plotting the Lorenz Curve, we compared income inequality between obese men and obese women. Several findings of this study need specific attention in addressing obesity. In the following paragraphs, we discuss more about these findings and policy recommendations.

Gender differences. The GC showed that obese women suffer more from income-inequality than men (Figure 2, plot B), with a higher GC (0.377 vs. 0.333). Based on our results, treatment for income inequality should target women with a priority on obese women as the population that suffers more from income inequality.

For the second set of analysis, we examined the association between income measured by the PIR and obesity for men and women. The results of the Modified Poisson regression models demonstrated that higher income in men was positively associated with a higher prevalence of obesity, but income in women was negatively associated with the prevalence of obesity in the highest income women. Men in higher income groups experienced higher probability of being obese, which may be explained by lower physical activities. The pattern in women was different. Other studies in the US [33] and out of, Canada [34] Sweden [35] and Ireland [36] concluded similar results, where the overweight and obesity prevalence rate was higher among the wealthy [36] population or was concentrated in wealthy people [37].

The stratified analysis of income across sex indicated that men in middle- and upper-income quintiles were more likely to be obese. The results for women were different. Results of adjusted models suggested that sociodemographic factors did not change the association between income and obesity in men except for the very top quintiles, but they played a role in women.

Cost of obesity. By considering $147 billion to nearly $210 billion as an annual estimated cost of obesity in the US [15,16] and by recognizing the positive association between obesity and hypertension, type II diabetes and other diseases including cardiovascular problems [38], implementing policies to reduce the prevalence of obesity can save tax payers money. Policymakers should consider the different strategies for men and women when obtaining obesity-reducing policies [35].

Physical activities and obesity. By highlighting the importance of physical activities and another aspect of this study that needs attention—the association between obesity and physical activities—our findings showed that men and women that did not partake in vigorous or moderate activity were more likely to be obese. In a study published by Harmon (2014), association between obesity and the natural environment (higher obesity in counties with hot summers or cold winters) was mediated by physical activity; these findings emphasized the importance of physical activities [34,39]. Any strategies to promote physical activity may help mitigate the obesity prevalence in men and women.

Addressing fundamental inequality. Furthermore, obesity is problematic for vulnerable groups such as communities of color [40] and women because of structural inequality such as an income gap [2,3], occupational risk factors [41] and wage disparities [42]. Our subset analysis on GC showed that obese Black Non-Hispanics with GC 0.380 (SE:0.004) experienced higher income inequality than White-NH (GC 0.301, SE: 0.004). In developing obesity-reducing policies, addressing these fundamental inequalities need essential attention [38]. Additionally, developing nutritional intervention programs such as community CSAs—with prioritization of low-income communities—may improve access to healthy foods with lower cost [43], reduce obesity, and ensure income equity in low-income communities.

Conducting more research focusing on gender differences. In spite of a fair amount of research regarding the impact of income inequality and health, [44,45,46,47] and some studies suggesting that income inequality is not associated with individual health outcomes [48,49,50], little is known about the impact of income inequality and obesity on men and women. We found only a few articles to address income inequality in US adults. One of those articles by Campbell et al. (2019) reported that income was negatively associated with weight status for men who were highly socially integrated, but that income was positively associated with weight status among men who had lower social integration [21]. Kim et al. (2018) reported that income inequality and a lower poverty percentage were significantly associated with lower obesity rates in men [51].

Most of the published articles on obesity and sex differences discussed the impact of geographical location [52], neighborhood factors [53], behavioral and physical activities [54,55] and income inequality [56,57]. It is essential to conduct more research regarding the differences between men’s and women’s health, with a focus on income differences and income inequality.

Several aspects of the present study deserve comment. The data were cross-sectional; therefore, we could not rule out the possibility of reverse causation. The evidence indicated that the extent of bias due to reverse causation was largely indirect [58]. The NHANES data had some limitations regarding the income variable and did not report real income; instead, income was reported as a categorical variable. Employing household income as a continuous variable could provide us a better opportunity to find the impact of income differences instead of a proxy variable such as the PIR. Another potential limitation was the BMI. In this study, using the CDC approach, we defined obesity if the BMI was ≥30 kg/m^2^ as documented by the CDC “BMI is screening tool, and it does not diagnose body fatness or health.” Additionally, to avoid any bias in treatment, appropriate assessments should be performed by a trained health care provider to evaluate an individual’s health status and risks [59].

There are also strengths to this study. To our knowledge, this is the first study to examine the relationship of income inequality and obesity in a wide range of the NHANES data (1999–2016) and by sex. In addition, we used weighted models that made our findings nationally representative estimates, increasing the generalizability of these results. We must note that the Lorenz curves are unaffected by the mean of the distribution, and “they cannot be used to rank distributions in terms of social welfare, only in terms of inequality [60].” To capture more variances, we should control our models based on the geographical variables such as urban and rural areas and neighborhoods or at the county level—the next step in our study.

## 5. Conclusions

The association between the PIR and obesity operates differently for men and women. Our findings showed that individuals living in an economically diverse vs. an economically homogeneous geographic area were going to have an impact on obesity that was independent of the direct effect of the individual’s family income on obesity. The PIR was positively associated with the prevalence of obesity in men in the higher quintile of PIR but negatively associated with women’s obesity in the same group. Our findings also indicate that income inequality plays different roles between men and women—obese women experienced higher income inequality than men. Policymakers should consider a combination of local and federal policies similar to farm bill policies [61] as short- and long-term strategies to improve health outcomes and to better distribute the income. Specifically, for low-income, vulnerable communities, the resources need to combat obesity through access to fresh foods and CSA programs as nutritional interventions and not only promote access to healthy foods but also generate higher income for low-income communities and reduce income inequalities as a long-term sustainable strategy. Long-term, multi-level, community-level interventions should reduce obesity, specifically for men in all levels of income. In women, the intervention should prioritize lower-income women.

## Figures and Tables

**Figure 1 ijerph-18-07079-f001:**
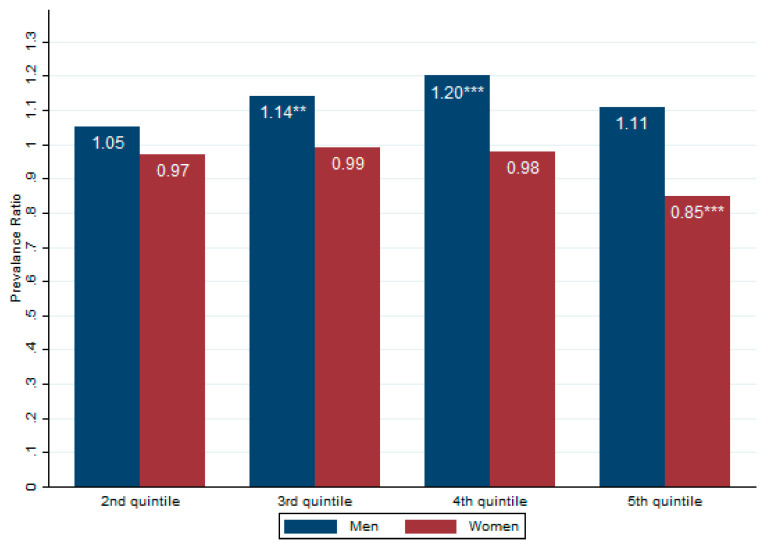
Association between income inequality and obesity in US men and women. Notes: Values show the prevalence ratio for adjusted model 4 and 6 (see Table 3 for more details. ** *p* < 0.01, *** *p* < 0.001).

**Figure 2 ijerph-18-07079-f002:**
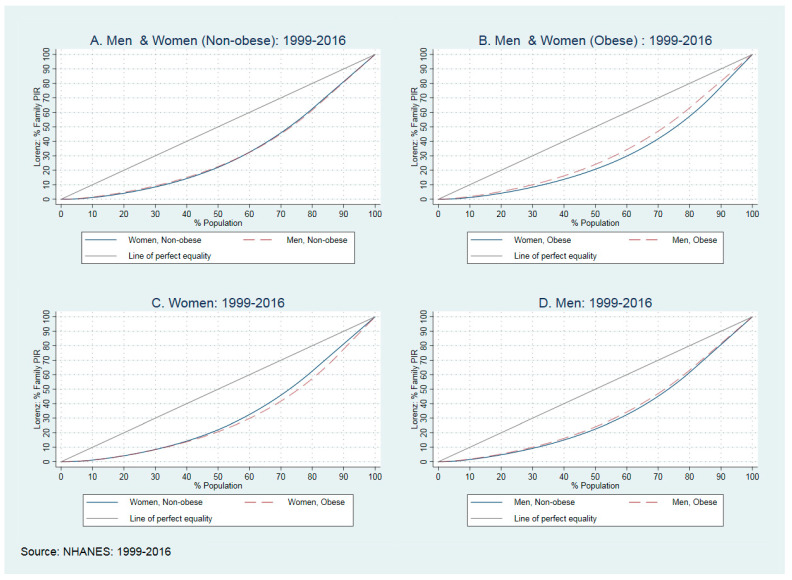
The Lorenz curves and Gini coefficients in men and women between 1999–2016.

**Table 1 ijerph-18-07079-t001:** Comparing study sample characteristics between men and women, US Adults over 20 years of age in 1999–2016 National Health and Nutrition Examination Survey (N = 36,665).

	Men	Women	All Participants	*p*-Value
Obese (%)	33.1	36.6	34.9	*p* < 0.001
**Ratio of family income to poverty (%)**				
1st quintile (PIR: 0.00–0.80)	8.4	10.3	9.3	*p* < 0.001
2nd quintile (PIR: 0.81–1.36)	11.9	13.7	12.9	*p* < 0.001
3rd quintile (PIR: 1.37–2.33)	17.0	18.4	17.7	*p* < 0.001
4th quintile (PIR:2.34–4.10)	25.2	24.1	24.6	*p* = 0.011
5th quintile (PIR:4.11–5.00)	37.4	33.4	35.4	*p* < 0.001
**Sociodemographic Variables**				
Age in years at screening (Mean, SD)	46.0 (18.0)	47.9 (17.9)	47.0 (18.0)	*p* < 0.001
Females (%)	-	-	50.6	
**Racial/Ethnic Groups (%)**				
White NH	70.8	71.0	70.9	*p* = 0.577
Black NH	9.8	11.2	10.5	*p* < 0.001
Mexican American	8.5	6.7	7.6	*p* < 0.001
Other	10.9	11.1	11.0	*p*= 0.606
**Marital Status (%)**				
Married	68.3	60.3	64.2	*p* < 0.001
**Education (%)**				
Less than high school	17.5	16.0	16.7	*p* < 0.001
High school graduate/GED	24.5	22.9	23.7	*p* = 0.004
More than high school	58.0	61.1	59.6	*p* < 0.001
**Health System Variables (%)**				
Covered by any kind of health insurance	79.9	84.7	82.3	*p* < 0.001
**Health behaviors**				
**Smoking Status (%)**				
Never	46.5	59.0	52.8	*p* < 0.001
Former	29.0	21.4	25.1	*p* < 0.001
Current	24.5	19.6	22.1	*p* < 0.001
**Drinking Status (%)**				
Never	7.6	17.4	12.6	*p* < 0.001
Former	7.1	15.7	11.5	*p* < 0.001
Current	85.2	67.0	76.0	*p* < 0.001
**Physical Inactivity (%)**				
Has No Rigorous or Moderate Activities	38.5	42.6	40.5	*p* < 0.001
**Self-reported Health (%)**				
Fair-poor (=1, if fair-poor)	15.8	17.5	16.7	*p* < 0.001
**HH Structure (%)**				
Live Alone (=1, if alone)	11.8	14.9	13.4	*p* < 0.001
Head of Household (Household reference person, Female)	72.3	43.7	57.8	*p* < 0.001

Notes: NH = Non-Hispanic; (1) We defined quintile based on the GC calculated from the ratio of family income to poverty; (2) Bonferroni correction (*p* < 0.002) shows the significant difference between men and women.

**Table 2 ijerph-18-07079-t002:** Association between income differences and obesity in US Adults in the1999–2016 National Health and Nutrition Examination Survey.

	All Participants			
	Unadjusted (Model 1)		Adjusted (Model 2)	
	PR	95%-CI	PR	95%-CI
**Ratio of family income to poverty (Ref. 1st quintile** **; PIR < 0.80)**				
2nd quintile (PIR: 0.81–1.36)	1.00	[0.94]–[1.06]	0.99	[0.93]–[1.05]
3rd quintile (PIR: 1.37–2.33)	1.01	[0.95]–[1.08]	1.02	[0.96]–[1.09]
4th quintile (PIR:2.34–4.10)	0.98	[0.93]–[1.04]	1.04	[0.98]–[1.11]
5th quintile (PIR:4.11–5.00)	0.85 ***	[0.80]–[0.91]	0.94	[0.87]–[1.01]
**Sociodemographic Variables**				
Female (Ref. Female)			1.04 *	[1.01]–[1.08]
Age in years at screening (Mean, SE)			1.00	[1.00]–[1.00]
**Racial/Ethnical Groups** (Ref. White NH)				
Black NH			1.28 ***	[1.23]–[1.34]
Mexican American			1.11 ***	[1.05]–[1.18]
Other			0.83 ***	[0.77]–[0.89]
**Marital Status** (Ref. Married)				
Married			1.07 **	[1.02]–[1.12]
**Education** (Ref. Less than high school)				
High school graduate/GED			1.17 ***	[1.11]–[1.23]
More than high school			1.10 ***	[1.04]–[1.15]
**Health System Variables** (Ref. Has HI)				
Covered by any type of health insurance			1.09 **	[1.04]–[1.15]
**Health behaviors**				
**Smoking Status** (Ref. Never smoked)				
Former			1.08 **	[1.03]–[1.13]
Current			0.83 ***	[0.78]–[0.87]
**Drinking Status** (Ref. Never drink)				
Former			1.18 ***	[1.10]–[1.26]
Current			0.97	[0.92]–[1.03]
**Physical Inactivity** (Ref. No vigorous or moderate activities)				
Has No Vigorous or Moderate Activities			1.27 ***	[1.23]–[1.32]
**Self-reported Health** (Ref. fair-poor)				
Fair-poor			1.42 ***	[1.36]–[1.48]
**Household Structure**				
Live Alone (=1, if alone)			0.99	[0.93]–[1.06]
Head of Household (Household reference person, Female)			0.96	[0.92]–[1.00]
N	36,665		36,665	

* *p* < 0.05, ** *p* < 0.01, *** *p* < 0.001 PR = prevalence ratio. Notes: (1) We defined quintile based on the GC calculated from ratio of family income to poverty. (2) Variance in the number of observations is due to some missing data.

**Table 3 ijerph-18-07079-t003:** Association between income differences and obesity in US Adults in the 1999–2016 National Health and Nutrition Examination Survey.

	Men				Women			
	Unadjusted (Model 3)		Adjusted (Model 4)		Unadjusted (Model 5)		Adjusted (Model 6)	
	PR	95%-CI	PR	95%-CI	PR	95%-CI	PR	95%-CI
**Ratio of family income to poverty (Ref. 1st quintile** **; PIR < 0.80)**								
2nd quintile (PIR: 0.81–1.36)	1.09	[0.97]–[1.22]	1.05	[0.94]–[1.17]	0.96	[0.89]–[1.03]	0.97	[0.90]–[1.05]
3rd quintile (PIR: 1.37–2.33)	1.22 ***	[1.10]–[1.34]	1.14 **	[1.03]–[1.26]	0.92 *	[0.85]–[0.99]	0.99	[0.91]–[1.06]
4th quintile (PIR:2.34–4.10)	1.26 ***	[1.15]–[1.39]	1.20 ***	[1.09]–[1.34]	0.84 ***	[0.78]–[0.91]	0.98	[0.90]–[1.05]
5th quintile (PIR:4.11–5.00)	1.16 **	[1.05]–[1.29]	1.11	[0.99]–[1.24]	0.68 ***	[0.63]–[0.74]	0.85 ***	[0.77]–[0.93]
**Sociodemographic Variables**								
Age in years at screening (Mean, SE)			1.00	[1.00]–[1.00]			1.00	[1.00]–[1.00]
**Racial/Ethnic Groups (Ref. White NH)**								
Black NH			1.11 **	[1.04]–[1.19]			1.41 ***	[1.34]–[1.49]
Mexican American			1.11 *	[1.02]–[1.22]			1.12 **	[1.04]–[1.21]
Other			0.86 **	[0.78]–[0.94]			0.81 ***	[0.74]–[0.89]
**Marital Status**								
Married			1.18***	[1.10]–[1.27]			1.03	[0.97]–[1.09]
**Education**								
High school graduate/GED			1.21 ***	[1.11]–[1.32]			1.13 ***	[1.06]–[1.21]
More than high school			1.16 **	[1.06]–[1.27]			1.03	[0.97]–[1.10]
**Health System Variables**								
Covered by any type of health insurance			1.13 **	[1.04]–[1.23]			1.03	[0.97]–[1.09]
**Health behaviors**								
**Smoking Status (Ref. Never smoked)**								
Former			1.08 *	[1.01]–[1.14]			1.08 *	[1.01]–[1.15]
Current			0.78 ***	[0.72]–[0.85]			0.88 **	[0.82]–[0.95]
**Drinking Status (Ref. Never drank)**								
Former			1.25 ***	[1.12]–[1.40]			1.16 ***	[1.08]–[1.25]
Current			1.04	[0.94]–[1.14]			0.96	[0.90]–[1.03]
**Physical Inactivity**								
Has No Vigorous or Moderate Activities			1.26 ***	[1.18]–[1.34]			1.27 ***	[1.21]–[1.34]
**Self-report Health**								
Fair-poor (=1, if fair-poor)			1.36 ***	[1.27]–[1.46]			1.45 ***	[1.38]–[1.52]
**Household Structure**								
Live Alone (=1, if alone)			1.09	[0.98]–[1.20]			0.95	[0.88]–[1.03]
Head of Household (Household reference person, Female)			1.02	[0.95]–[1.09]			0.95	[0.90]–[1.00]
N	18,518		18,518		18,147		18,147	

* *p* < 0.05, ** *p* < 0.01, *** *p* < 0.001. PR = prevalence ratio. Note. Variance in the number of observations is due to some missing data.

## Data Availability

The data presented in this study are openly available in [National Health and Nutrition Examination Survey (NHANES) at https://www.cdc.gov/nchs/nhanes/index.htm] (accessed on 11 February 2020).

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
