# Peer review of "Income Inequality and Obesity among US Adults 1999–2016: Does Sex Matter?"

_ijerph, 2021, doi:10.3390/ijerph18137079_

Round 1
Reviewer 1 Report
The manuscript presented by Zare et al., " Income Inequality and Obesity Among US Adults 1999-2016: Does Sex Matter?" intended to demonstrate the association between income inequality and obesity, focusing on the differences between women and men.
Overall, the study observed that the association between income inequality and obesity is different for men and women. The higher income inequality was positively associated with the prevalence of obesity in men but negatively associated with women’s obesity in the same groups.
In order to clarify some points, I would like to suggest some amendments.
- Since muscle weighs more than fat, BMI isn't accurate in patients with high amounts of muscle. A fit person with dense muscle could be categorized as overweight based on BMI. It is important to consider that.
- Please, include the statistical test used for each table in the table legend, not only the p value. Legends are incomplete
- Figure legend 1 is incomplete. What do the stars stand for?
- I would divide the result sections using subtitles in order to make the results more clear. There is only one subtitle and all the analysis after that. I would divide by different analysis model, always giving a brief introduction about why that kind of analysis was done.
- It would be interesting to include in the discussion since the authors already mention why geographical variables should be used to control the models. Would the climate (cold vs more tropical) have an impact?
- A discussion about ethnic group/ women vs men according to the Gini Coefficient Categories and obesity would be interesting since genetic factors also might be involved.
Reviewer 2 Report
This manuscript is very interesting and it is about a study that looks at obesity and income inequality by sex and race/VM status, among other things. The subject matter is important. The manuscript is also written and articulated well. However, there is some information that I found was missing that could otherwise strengthen the paper further. The manuscript requires major changes as indicated below:
- Line 13 – perhaps add “yet” after the word “sex” and replace the semicolon ; with a comma.
- Lines 14 to 15 – where was the study conducted? Later in the introduction you indicate the USA but it would be helpful to state this in the abstract as this is an international journal.
- Lines 25 to 26 – Seems a bit abrupt…Perhaps replace “Policymakers should consider” with the following “Our results suggest that policymakers should consider…”
- Lines 48 to 49 – This sentence needs to be modified. Currently it states that lower income populations have higher rates of behavioral risk factors…but risk factors for what? Obesity? Poverty? This needs to be clear.
- Lines 72 to 73 – The literature thus far is good but a question that remains is: why should we care about obesity? The answers are in the literature, thus, the paper can benefit immensely from the literature about how obesity is problematic for health and wellbeing. It may be worthwhile to cite the work of researchers who discuss this. For example, obesity has economic costs of $3.38 to $6.38 billion in the USA (See Trogdon, J.G.; Finkelstein, E.A.; Hylands, T.; Dellea, P.S.; Kamal-Bahl, S.J. Indirect costs of obesity: A review of the current literature. Obes. Rev. 2008, 9, 489–500. "Obesity is also considered unhealthy because it is linked to hypertension, type II diabetes, cardiovascular problems, gallbladder disease, certain arthritic conditions, and cancer", see https://www.mdpi.com/2075-4698/9/3/59/htm ).
- Line 113 – what is the range of incomes for each of these quintiles?
- Line 128 – Are female headed households supposed to be a proxy for single mothers? If so, please state this.
- Line 153 – This sentence could be better stated. Perhaps replace “men were in the top quintiles” with “More men were in the top quintiles than women”.
- Line 178 – the “q” in quintile is a different font/size. Please revise.
- Line 254 – has extra spaces; please revise.
- Line 281 to 285 – The literature here is a bit scant and could benefit enormously from the work of particular researchers. For example, it may be worthwhile to cite the work of researchers who state that high BMI is problematic for racialized groups (see Chiu, M.; Austin, P.C.; Manuel, D.G.; Shah, B.R.; Tu, J.V. Deriving ethnic-specific BMI cutoff points for assessing diabetes risk. Diabetes Care 2011, 34, 1741–1748); Furthermore, obesity is problematic for vulnerable groups such as racialized minorities and women because of structural factors and inequality in society, including occupational risk factors. For example, “the underlying issues of income inequality, social exclusion, sexism, and racism need to be addressed” because “structural elements of inequality in society, account for obesity-related mortality differentials among people” see full citation: Syed, Iffath U. 2019. "In Biomedicine, Thin Is Still In: Obesity Surveillance among Racialized, (Im)migrant, and Female Bodies" Societies 9, no. 3: 59. https://doi.org/10.3390/soc9030059
- Line 388 of references should have regular font, and not all caps.
If you decide to incorporate these revisions, please upload a manuscript that contains tracked changes or other method to highlight revisions. Thank you for the opportunity to review this work.
Round 2
Reviewer 2 Report
This manuscript is very interesting and it is about a study that looks at obesity and income inequality by sex and race/VM status, among other things. The subject matter is important. The manuscript is also written and articulated well. The authors followed the reviewer’s suggestions for revision and many of the revisions have been highlighted in red font throughout. I only have two minor suggestions. In line 36 it states “Despite of a little progress,”; this may not be grammatically correct; perhaps revise to “Despite little progress,”. In line 107, coefficient should have the first letter capitalized, so it is Gini Coefficient (GC) instead of ‘coefficient’. It is recommended that the manuscript should be published. Thank you for the opportunity to review this work.